# A Distance Covariance-based Kernel for Nonlinear Causal Clustering in Heterogeneous Populations

## Abstract

We consider the problem of causal structure learning in the setting of heterogeneous populations, i.e., populations in which a single causal structure does not adequately represent all population members, as is common in biological and social sciences. To this end, we introduce a distance covariance-based kernel designed specifically to measure the similarity between the underlying nonlinear causal structures of different samples. This kernel enables us to perform clustering to identify the homogeneous subpopulations. Indeed, we prove the corresponding feature map is a statistically consistent estimator of nonlinear independence structure, rendering the kernel itself a statistical test for the hypothesis that sets of samples come from different generating causal structures. We can then use existing methods to learn a causal structure for each of these subpopulations. We demonstrate using our kernel for causal clustering with an application in genetics, allowing us to reason about the latent transcription factor networks regulating measured gene expression levels.

## 1 Introduction

Learning causal relationships from observational and experimental data is one of the fundamental goals of scientific research, and causal inference methods are thus used in a wide variety of fields. The resulting variety of applications nevertheless share some common difficulties, such as causal inference from complex time-series data (Eichler, 2012) or the underlying causal structure being obscured by unmeasured confounders (Greenland et al., 1999). Another common difficulty, especially for applications in the biological and social sciences, is causal inference from heterogeneous populations (Xie, 2013; Brand and Thomas, 2013)—addressing this difficulty is our main motivation.

In general terms, we understand a heterogeneous population to be one whose members are not adequately described by a single model but rather better described by a collection of models. Within our context of causal structure learning, this means a population is heterogeneous if some samples are generated by different causal structures—we call this *structural* heterogeneity. We note that there are other kinds of heterogeneity, such as that in samples generated by different joint distributions over the same causal structure, which are not the scope of this work.

A specific example of structural heterogeneity can be found in genetics: causal methods are used to learn the structure of gene regulatory networks (Emmert-Streib et al., 2012), and gene expression data from a single recording or experiment may include thousands of genes, many of which are involved in entirely different networks (Liu, 2015); thus, attempting to learn a single causal structure for all of the genes will obscure the fact that different sets of them have different structures.

The bulk of our work in this paper, and our main contribution, is to introduce the *dependence contribution kernel*, which facilitates a flexible and easily extensible approach to causal clustering: first perform clustering to identify structurally homogeneous subsets of samples, and then proceed with the actual learning task on each cluster. We prove that our kernel is a statistically consistent estimator of the similarity of the causal structures underlying different samples and can thus be used to find clusters that minimize structural heterogeneity for causal structure learning tasks. Furthermore, the kernel is derived from the distance covariance (Székely et al., 2007), imbuing it with the ability to detect nonlinear dependence. It can easily be used in a wide array of clustering algorithms, such as $k$-means, DBSCAN, spectral clustering, or any other method that analogously makes use of a similarity (or distance) measure between samples (Filippone et al., 2008).

The rest of the paper is organized as follows: We finish this section by discussing some of the most relevant related work from the causal inference and statistics literature. All of Section 2 is devoted to the theory underlying our dependence contribution kernel, including a comparison of the familiar product-moment covariance with the distance covariance (Section 2.1), defining an equivalence class of causal models with a convenient representation in the kernel space (Section 2.2), and the actual definition of our kernel and proofs of its relevant properties (Section 2.3). Next, in Section 3, we demonstrate causal clustering with the kernel on a heterogeneous gene expression data set, finding structurally homogeneous clusters for which we then learn latent causal measurement models, allowing us to reason about the different transcription factor networks responsible for regulating the measured gene expression levels. Finally, we conclude in Section 4 mentioning possible future work.

## 1.1 Related Work

Causal inference in heterogeneous populations sometimes refers to data-fusion (Bareinboim and Pearl, 2016), i.e., combining known homogeneous subpopulations and performing causal inference on the resulting heterogeneous population, or similarly, it can refer to meta-learning using known subpopulations (Sharma et al., 2019). Other times, it refers to estimating heterogeneous treatment effects (Xie et al., 2012; Athey and Imbens, 2015). However, in our case, the subpopulations are not known and we rather consider the problem of learning which samples come from which subpopulation, and these are differentiated according to structure instead of treatment effect.

Previous work on causal clustering has focused more on the causal modeling aspect, using stronger assumptions about the underlying structures to learn more detailed models. For example, Kummerfeld et al. (2014); Kummerfeld and Ramsey (2016) focus on causal clustering in measurement models, with the goal of clustering different features together to study their latent causal structure, based on tetrad constraints within the linear product-moment covariance matrix. Huang and Zhang (2019) define a class of causal models facilitating mechanism-based clustering, learning causal models both for clusters of samples as well as a shared one for all samples, assuming the underlying structures are linear non-Gaussian. Saeed et al. (2020) characterize distributions arising from mixtures of directed acyclic graph (DAG) causal models (i.e., causal models without latent or selection variables), trying to learn both the component DAGs and a representation of how they are mixed. All of these approaches, like most causal inference methods, make specific (and for some applications, restrictive) assumptions about the underlying distributions or causal structures.

In contrast, our method is not tied to specific distributional assumptions such as linearity or (non)Gaussianity—we assume there are enough samples for statistical inference, as well as the usual causal Markov and faithfulness assumptions. For the first step, we cluster samples together if they (implicitly, in the kernel space) have similar nonlinear independence structures. For the second step, causal structure learning, any existing method (along with its corresponding assumptions) can in principle be used. In our gene expression data application (Section 3), the measurement dependence inducing latent (MeDIL) causal model framework (Markham and Grosse-Wentrup, 2020), which assumes the data consists of measurement variables that are causally connected only through latent variables, seems appropriate, however other applications can easily use other methods. For example, component and mixture DAGs (Saeed et al., 2020) can be better learned when one first knows which samples come from which component—clustering with our kernel ensures samples in different

85 clusters come from different DAGs, and so using their method instead of the MeDIL framework
86 would be a natural choice for applications in which a DAG (without any latents) is more appropriate.

## 2 Theory

### 2.1 Product-moment Covariance, Distance Covariance, and Dependence Contribution

89 Though there is more to causal relationships than probabilistic dependence, causal inference methods
90 based on graphical models ultimately rely on at least implicitly learning conditional independence
91 (CI) relations. CI relations can be estimated in many ways, with different dependence measures
92 and tests each having their own theoretical guarantees and being better suited for distributions of
93 various different kinds of data (e.g., categorical, discrete, or continuous) and with various kinds
94 of relationships (e.g., linear, monotonic nonlinear, arbitrary nonlinear) and with different testing
95 assumptions (see Tjøstheim et al., 2018, for a comprehensive overview).

96 A widely used measure of dependence is the *product-moment covariance*, often just called covariance,
97 which is defined for two zero-mean random variables $X_1$ and $X_2$ as the scalar value $\mathrm{cov}(X_1, X_2) =$
98 $\mathrm{E}[X_1 X_2]$. This can be extended from a pair of random variables to every pair of variables in a
99 random vector, thus returning a matrix instead of a scalar. The covariance matrix for a vector of
100 zero-mean random variables $\mathbf{X} = (X_1, \ldots, X_m)$ can be estimated from a set $S \in \mathbb{R}^{n,m}$ of $n$ samples
101 as $\hat{\Sigma}_{\mathbf{X}} = \frac{1}{n} S^\top S$, and the $j, j'$-th value of $\hat{\Sigma}_{\mathbf{X}}$ is thus the estimate $\hat{\mathrm{cov}}(X_j, X'_j)$.

102 Two random variables being probabilistically independent (denoted $\perp\!\!\!\perp$) implies that their product-
103 moment covariance is zero, i.e., $X_j \perp\!\!\!\perp X_{j'} \implies \mathrm{cov}(X_j, X_{j'}) = 0$ (importantly, the inverse of this
104 does not hold). Thus, the estimated product-moment covariance can be used in statistical hypothesis
105 testing for probabilistic independence (Wasserman, 2013, Ch. 10): $X_j$ and $X_{j'}$ are assumed to
106 be independent if and only if $\hat{\mathrm{cov}}(X_j, X_{j'})$ is sufficiently close to 0. However, this method has
107 an important problem: the product-moment covariance is only a valid test statistic against *linear*
108 dependence.

109 Székely et al. (2007) introduce the *distance covariance* to remedy this problem: random variables are
110 probabilistically independent if and only if their distance covariance is zero, i.e., $X_j \perp\!\!\!\perp X_{j'} \iff$
111 $\mathrm{dCov}(X_j, X_{j'}) = 0$, resulting in the estimated distance covariance being a valid test statistic against
112 all types of dependence. The distance covariance is related to the product-moment covariance by
113 $\mathrm{dCov}^2(X_j, X_{j'}) = \mathrm{cov}(|X_j - X'_j|, |X_{j'} - X'_{j'}|) - 2\mathrm{cov}(|X_j - X'_j|, |X_{j'} - X''_{j'}|)$, where $(X'_j, X'_{j'})$
114 and $(X''_j, X''_{j'})$ are independent and identically distributed (iid) copies of $(X_j, X_{j'})$ (Székely and
115 Rizzo, 2014). The key intuition here is that the distances (e.g., $|X_j - X'_j|$) constitute a nonlinear
116 projection, so that using the linear product-moment covariance in this projected space allows for the
117 detection of nonlinear dependence in the original space.

118 Note that $\mathrm{dCov}$ is typically defined to be a scalar value when taken between two arbitrary-dimensional
119 random vectors, but our restricted presentation of it above in terms of random variables is to make
120 it more obviously analogous to the product-moment covariance between random variables. Thus,
121 corresponding to $\hat{\Sigma}_{\mathbf{X}}$ for random vectors, we define the following:

122 **Definition 1** Let $S \in \mathbb{R}^{n,m}$ be a set of $n$ samples from the vector of random variables $\mathbf{X} =$
123 $(X_1, \ldots, X_m)$. For each $j \in \{1, \ldots, m\}$ and $i, i' \in \{1, \ldots, n\}$, define the pairwise distance matrix
124 $D^j$, with values given by $D^j_{i,i'} := |S_{i,j} - S_{i',j}|$. Now define the corresponding doubly-centered
125 matrices $C^j_{i,i'} := D^j_{i,i'} - \bar{D}^j_{i,\cdot} - \bar{D}^j_{\cdot,i'} + \bar{D}^j_{\cdot,\cdot}$, where putting a bar over the matrix and replacing
126 an index $i$ or $i'$ with $\cdot$ denotes taking the mean over that index. Define the matrix $L \in \mathbb{R}^{n^2,m}$ so
127 that each column is a flattened doubly-centered distance matrix, $L := (\mathrm{vec}(C^1), \ldots, \mathrm{vec}(C^m))$,
128 where $\mathrm{vec}(C^j)$ denotes "flattening" matrix $C^j$ into a column vector. Finally, the estimated *distance*
129 *covariance matrix* over sample $S$ is defined as $\hat{\Delta}_{\mathbf{X}} := \frac{1}{n^2} L^\top L$.

130 Analogous to $\hat{\Sigma}_{\mathbf{X}}$, the $j, j'$-th entry of $\hat{\Delta}_{\mathbf{X}}$ corresponds to $\mathrm{d\hat{C}ov}^2(X_j, X_{j'})$—indeed it is mathemati-
131 cally equivalent to computing each pairwise distance covariance value and then manually filling in

the matter. The novelty of our Definition 1 is in finding a matrix of pairwise values instead of a single value for the distance covariance between random vectors, which helps provide an intuition for our next definition:

**Definition 2** Let $S \in \mathbb{R}^{n,m}$ be a set of $n$ samples from the vector of random variables $\mathbf{X} = (X_1, \ldots, X_m)$; note that we consistently use indices $i, i' \in \{1, \ldots, n\}$ and $j, j' \in \{1, \ldots, m\}$. Let $D \in \mathbb{R}^{n,n,m}$ denote the 3-dimensional array of stacked pairwise distance matrices defined by $D_{i,i',j} := |S_{i,j} - S_{i',j}|$, and use $C \in \mathbb{R}^{n,n,m}$ to denote these same distance matrices after being doubly-centered, i.e., $C_{i,i',j} := D_{i,i',j} - \bar{D}_{i,\cdot,j} - \bar{D}_{\cdot,i',j} + \bar{D}_{\cdot,\cdot,j}$, where replacing an index $i$ or $i'$ with $\cdot$ denotes the entire (lower-dimensional) subarray over that index, and writing a bar, $\bar{D}$, denotes taking the mean over that subarray. Then standardize the doubly-centered distances to get $Z_{i,i',j} := \frac{C_{i,i',j}}{\bar{D}_{\cdot,\cdot,j}}$. Finally, the *dependence contribution map*, $\varphi : \mathbb{R}^m \to \mathbb{R}^{m,m}$, is defined as

$$\varphi(S_{i,\cdot}) := Z_{i,\cdot,\cdot}^\top Z_{i,\cdot,\cdot} - \mathcal{T}(\alpha),$$

where $\mathcal{T}(\alpha) \in \mathbb{R}^{m,m}$ is a matrix of scaled critical values corresponding to a given significance level $\alpha$ with zeros along the diagonal, i.e., $\mathcal{T}(\alpha)_{j,j'} = \begin{cases} 0, & \text{if } j = j' \\ \frac{1}{n}\chi^2_{1-\alpha}(1), & \text{otherwise} \end{cases}$, with $\chi^2_{1-\alpha}(1)$ being the $1 - \alpha$ quantile of the chi-square distribution with 1 degree of freedom.

Notice the similarity between Definitions 2 and 1: if we set $\mathcal{T}(\alpha)$ to be a matrix of 0s and forgo standardization (i.e., use $C$ instead of $Z$), then $\frac{1}{n^2}\sum_{i=1}^n \varphi(S_{i,\cdot}) = \hat{\Delta}_{\mathbf{X}}$. Now, the differences: $\hat{\Delta}_{\mathbf{X}}$ is a single matrix computed over an entire set of samples, whereas $\varphi$ is a map that projects each given sample to a new feature space; each entry of $\hat{\Delta}_{\mathbf{X}}$ is simply a distance covariance value, whereas each entry of the sum of $\varphi(S_{i,\cdot})$ over $i$, by using standardization (using $Z$ instead of $C$) and subtracting a critical value, corresponds to the result of using a distance covariance value in a statistical hypothesis test for independence—indeed:

**Lemma 3** Let $S \in \mathbb{R}^{n,m}$ be a set of $n$ iid samples from random variables $X_1, \ldots, X_m$ with finite first moments. For a given significance level $\alpha$, under the null hypothesis of $X_j \perp\!\!\!\perp X_{j'}$, the test

$$\text{reject } h_\emptyset \text{ if } \quad \big( \sum_{i=1}^n \varphi(S_{i,\cdot}) \big)_{j,j'} > 0$$

is statistically consistent against all types of dependence.

*Proof.* This follows from (Székely and Rizzo, 2009, Theorem 5 and Corollary 2) and how $\varphi$ is defined to correspond to the difference between distance covariance and critical values. $\square$

These differences between $\hat{\Delta}_{\mathbf{X}}$ and $\varphi$ serve two important purposes: first, they ensure $\varphi$ maps to a Hilbert space so that our Definition 9 is a corresponding kernel function (Schölkopf et al., 2001); and second, as the name "dependence contribution map" suggests, they ensure $\varphi(S_{i,\cdot})$ is informative not just about distance covariance but about nonlinear dependence and about how the inclusion of sample $S_{i,\cdot}$ in a set of samples $S$ contributes to the dependence patterns estimated from $S$— this is the key intuition behind how our kernel function is used to learn structurally homogeneous sample subsets, as explicated in the following sections.

## 2.2 Causal Graphs in Kernel Space

In general, a full causal structure can only be learned with sufficient data about the effects of interventions, and thus causal structure learning from purely observational data is usually possible only up to an equivalence class of causal graphs (Spirtes et al., 2000; Pearl, 2009). For example, the classic PC and IC algorithms, under the assumptions of no selection bias and no confounding by latent variables, do not necessarily return a fully-specified DAG but instead return a mixed graph, containing possibly directed and undirected edges, representing the Markov equivalence class (Spirtes and Glymour, 1991; Pearl and Verma, 1995).

We now define a set of equivalence classes for ancestral graphs (AGs), which—unlike causal DAGs— do not assume the absence of selection bias and latent confounders (Richardson et al., 2002):

**Definition 4** Consider an arbitrary ancestral graph $\mathcal{A}$ with the set of vertices $V^{\mathcal{A}}$ and edge function $E^{\mathcal{A}}$, and denote the set of unconditional $m$-connection statements entailed by their corresponding unique maximal ancestral graph as $M^{\mathcal{A}} = \{(j, j') : j \not\perp_m j' \mid \emptyset\} \subseteq V^{\mathcal{A}} \times V^{\mathcal{A}}$. For any ancestral graph $\mathcal{A}'$ such that $V^{\mathcal{A}'} = V^{\mathcal{A}}$, define the *unconditional equivalence* relation denoted by '$\sim_{\mathrm{U}}$' as

$$\mathcal{A} \sim_{\mathrm{U}} \mathcal{A}' \quad \text{if and only if} \quad M^{\mathcal{A}} = M^{\mathcal{A}'}.$$

**Lemma 5** This lemma has two parts: (i) the relation $\sim_{\mathrm{U}}$ is an equivalence relation over the set of ancestral graphs $\mathbb{A}$; (ii) for an arbitrary ancestral graph $\mathcal{A} \in \mathbb{A}$, the bidirected graph $\mathcal{U}^{\mathcal{A}} = (V^{\mathcal{A}}, E^{\mathcal{U}})$, where $E^{\mathcal{U}}$ maps all pairs $(j, j') \in M^{\mathcal{A}}$ to the bidirected edge symbol '$\leftrightarrow$', is a unique *representative* of the equivalence class $[\mathcal{A}]$.

*Proof.* For (i), recall that an equivalence relation is any relation satisfying reflexivity, symmetry, and transitivity (Devlin, 2003), all of which are satisfied by $\sim_{\mathrm{U}}$ because of its correspondence to the relation '=' between sets. Thus, to prove (ii), it suffices to show that the map $s : \mathbb{A}/\sim_{\mathrm{U}} \to \mathbb{A}, [\mathcal{A}] \mapsto \mathcal{U}^A$ is injective (i.e, that it is a *section*) and that $[s([\mathcal{A}])] = [A]$ (Mac Lane, 2013). The key to the proof is the observation that $\mathcal{U}^{\mathcal{A}}$, because it contains only bidirected edges, is maximal and therefore entails exactly the unconditional $m$-separation statements $M^{\mathcal{A}}$, thus by (i) we have $\mathcal{U}^{\mathcal{A}} \sim_{\mathrm{U}} \mathcal{A}$ or equivalently $\mathcal{U}^{\mathcal{A}} \in [\mathcal{A}]$ or equivalently $[\mathcal{U}^{\mathcal{A}}] = [\mathcal{A}]$. Let $\mathcal{A}, \mathcal{A}'$ be arbitrary AGs, and assume $s([\mathcal{A}]) = s([\mathcal{A}'])$. Then by definition of $s$ we have $\mathcal{U}^{\mathcal{A}} = \mathcal{U}^{\mathcal{A}'}$, and by the observation above, $\mathcal{U}^{\mathcal{A}} \in [\mathcal{A}']$ and thus $[\mathcal{A}] = [\mathcal{A}']$, making $s$ injective. And finally, by the definition of $s$ and also by the observation above, $[s([\mathcal{A}])] = [\mathcal{U}^A] = [A]$, completing the proof. $\qquad\square$

This equivalence relation and its representatives has some important but perhaps subtle properties. First, it is different from Markov equivalence over AGs (which is characterized by partial ancestral graphs, PAGs) (Zhang, 2007)—it uses only unconditional $m$-separation while PAGs are learned from conditional $m$-separation statements. Second, because all DAGs are AGs, $\sim_{\mathrm{U}}$ is also an equivalence relation over DAGs. Third, being a representative means that every equivalence class includes exactly one fully bidirected graph (along with other equivalent AGs). Fourth, because each representative is formed by considering $m$-connected paths, $\mathcal{U}^{\mathcal{A}}$ is not equivalent to what would be generated by some "edge-wise" procedure, such as simply replacing every edge in a PAG/AG/DAG/Markov random field/moralized DAG with bidirected edges.Finally, its most important property is that it facilitates Theorem 8, for which we first need a few more definitions.

**Definition 6** Given arbitrary ancestral graphs $\mathcal{A}, \mathcal{A}' \in \mathbb{A}$ over the same set of vertices, define the *Hamming similarity product*, denoted '$\bullet$' as

$$\bullet : \mathbb{A} \times \mathbb{A} \to \mathbb{A} \quad \text{and} \quad \mathcal{A} \bullet \mathcal{A}' \mapsto \mathcal{H},$$

where $\mathcal{H} = (V^{\mathcal{A}}, E^{\mathcal{H}})$ and the function $E^{\mathcal{H}}(j, j') = $ '$\leftrightarrow$' if and only if $E^{\mathcal{A}}(j, j') = E^{\mathcal{A}'}(j, j')$.

In words, the Hamming similarity product between two ancestral graphs returns a fully bidirected graph, with edges only where the two graphs have the same edge type. Now, shifting from ancestral graphs to real-valued square matrices:

**Definition 7** Let '$\sim_{\mathrm{O}}$' denote the *orthant equivalence* relation ('orthant' is the generalization of 'quadrant' from $\mathbb{R}^2$ to arbitrarily higher dimensions) in square real matrices, i.e., for matrices $Y, Y' \in \mathbb{R}^{m,m}$ and with the element-wise function $\text{sign}(Y)_{j,j'} = \begin{cases} 1, & \text{if } Y_{j,j'} > 0 \text{ or } j = j', \\ -1, & \text{otherwise} \end{cases}$,

$$Y \sim_{\mathrm{O}} Y' \quad \text{if and only if} \quad \text{sign}(Y)_{j,j'} = \text{sign}(Y')_{j,j'} \text{ for all } j, j'.$$

**Theorem 8** Let $a$ be the map from the set of unconditional equivalence classes over ancestral graphs with $m$ vertices, $\mathbb{A}^m/\sim_{\mathrm{U}} = \mathbb{U}^m$, to the set of orthant equivalence classes over the image of $\varphi$, i.e., $m \times m$ symmetric real matrices with positive diagonal entries, $\varphi(\mathbb{R}^m)/\sim_{\mathrm{O}} = \mathbb{O}^m$, defined by $a : \mathcal{U} \mapsto O$, where $O_{j,j'} = \begin{cases} 1, & \text{if } E^{\mathcal{U}}(j, j') = \text{'}\leftrightarrow\text{'} \text{ or } j = j' \\ -1, & \text{otherwise} \end{cases}$. Then $a$ is a group isomorphism between $(\mathbb{U}^m, \bullet)$ and $(\mathbb{O}^m, \odot)$, where '$\odot$' denotes the element-wise product.

*Proof.* First, note that $(\mathbb{U}^m, \bullet)$ is indeed a group, satisfying the three group axioms (Artin, 2011): the representative of its identity element is the fully connected bidirected graph over $m$ vertices, $\mathcal{U}^{\mathbb{1}}$; each element is its own inverse; and $\bullet$ is associative. Likewise, $(\mathbb{O}^m, \odot)$ is a group with identity element $[\mathbb{1}^{m,m}]$, each element its own inverse, and the associative element-wise product operator.

Now, to show the two groups are isomorphic, it suffices to show (i) that $a$ is bijective and (ii) that for arbitrary $\mathcal{U}, \mathcal{U}' \in \mathbb{U}^m$, $a(\mathcal{U}) \odot a(\mathcal{U}') = a(\mathcal{U} \bullet \mathcal{U}')$. For (i) notice that if $U \neq U'$, then there must be at least one pair of vertices $j, j'$ such that $E^{\mathcal{U}}(j,j') \neq E^{\mathcal{U}'}(j,j')$ and thus clearly $O_{j,j'} \neq O'_{j,j'}$, so $a$ in injective. Furthermore, notice that every distinct $O \in \mathbb{O}^m$ is the image of some graph $\mathcal{U}$, so $a$ is also surjective. For (ii), for every $j, j' \in \{1, \ldots, m\}$, the definitions of $a$, $\odot$, and $\bullet$ ensure $a(\mathcal{U})_{j,j'} \odot a(\mathcal{U}')_{j,j'} = 1 \iff E^{\mathcal{U}}(j,j') = E^{\mathcal{U}'}(j,j') \iff 1 = a(\mathcal{U} \bullet \mathcal{U}')$, completing the proof. $\qquad\square$

For causal inference, which (often, but not necessarily) amounts to taking several samples in real space and inferring a single corresponding member in the space of ancestral graphs (or, more often, its quotient set by some equivalence relation), Theorem 8 means we can compare the different graphs of different sample sets without having to first move to the ancestral graph space.

Finally, notice the space of real square matrices is not a typical sample space but rather precisely (a superspace of) the space that our dependence contribution map $\varphi$ (Definition 2) maps samples to—this means that mapping samples with $\varphi$ allows us to make use of the group isomorphism. Though this already provides an intuition for why using $\varphi$ would help with causal clustering, explicitly mapping each sample with it would be unnecessarily computationally expensive, and we are ultimately interested in morphisms between *metric spaces* (not just groups) of samples and graphs. To address this, we thus now move on to defining a kernel for $\varphi$.

## 2.3 The Dependence Contribution Kernel

**Definition 9** Let $S, Z, \mathcal{T}$, and $\varphi$ be as in Definition 2. We define the *dependence contribution kernel* using the Frobenius (denoted by the subscript $_\mathrm{F}$) inner product and norm:

$$\kappa(S_{i,\cdot}, S_{i',\cdot}) = \frac{\langle \varphi(S_{i,\cdot}), \varphi(S_{i',\cdot}) \rangle_\mathrm{F}}{\|\varphi(S_{i,\cdot})\|_\mathrm{F} \|\varphi(S_{i',\cdot})\|_\mathrm{F}}$$

A more convenient expression for applying the kernel to a data set is obtained by first defining a helper kernel, $\gamma$ along with vec from Definition 1:

$$\gamma(S_{i,\cdot}, S_{i',\cdot}) = \langle \varphi(S_{i,\cdot}), \varphi(S_{i',\cdot}) \rangle_\mathrm{F}$$
$$= \left( (\mathrm{vec}(Z_{i,\cdot})^\top \mathrm{vec}(Z_{i',\cdot}))^2 - Z_{i,\cdot} \mathcal{T} Z_{i,\cdot}^\top - Z_{i',\cdot} \mathcal{T} Z_{i',\cdot}^\top + \|\mathcal{T}\|_2^2 \right)$$

This allows us to write

$$\kappa(s, s') = \frac{\gamma(S_{i,\cdot}, S_{i',\cdot})}{\gamma(S_{i,\cdot}, S_{i,\cdot})^{\frac{1}{2}} \gamma(S_{i',\cdot}, S_{i',\cdot})^{\frac{1}{2}}}$$

Finally, note that $\kappa$ can be readily implemented on an entire set of samples, returning an entire Gram (kernel) matrix instead of a scalar value, by replacing the matrix operations above with tensor operations and specifying the correct axes along which summation occurs—an implementation can be found in our open source Python package at `https://non-anonymous-link.after-review`.

A proper distance metric can also be obtained from this kernel through function composition: $\arccos \circ \kappa$. The key idea behind the kernel is that it is the cosine similarity in the space that $\varphi$ maps to, meaning for arbitrary sample points $x, x'$ it evaluates to $\cos(\theta)$, where $\theta$ is the angle between $\varphi(x)$ and $\varphi(x')$. In this space, $\theta$ represents the dissimilarity of the *dependence patterns* underlying $x$ and $x'$, without being biased by the possibly different magnitudes of $\varphi(x)$ and $\varphi(x')$ due to differing *variances*. Indeed, it can be used as a statistical test of whether samples come from different dependence structures and therefore causal models:

**Theorem 10** Let $S \in \mathbb{R}^{n,m}$, $S' \in \mathbb{R}^{n',m}$ be sets of $n, n'$ iid samples drawn respectively from the random variables $X = (X_1, \ldots, X_m)$ and $X' = (X'_1, \ldots, X'_m)$ with finite first moments. Then,

$$\sum_{i=1}^{n} \sum_{i'=1}^{n'} \kappa(S_{i,\cdot}, S'_{i',\cdot}) < 0 \implies \exists j, j' \in \{1, \ldots, m\} \text{ such that } \mathcal{I}(X_j, X_{j'}, \emptyset) \neq \mathcal{I}(X'_j, X'_{j'}, \emptyset).$$

*Proof.* Through Slutsky's Theorem (see Takeshi, 1985, Theorem 3.2.7) and the continuous mapping theorem (see Van der Vaart, 2000, Theorem 2.3), the consistency of $\varphi$ (Lemma 3) guarantees the consistency of $\kappa$. Because the numerator of $\kappa$ is a Frobenius inner product of $\varphi$,

$$\sum_{i=1}^{n} \sum_{i'=1}^{n'} \kappa(S_{i,\cdot}, S'_{i',\cdot}) \propto \sum_{i=1}^{n} \sum_{i'=1}^{n'} \sum_{j=1}^{m} \sum_{j'=1}^{m} \varphi(S_{i,\cdot})_{j,j'} \varphi(S'_{i',\cdot})_{j,j'}.$$

Thus, in order for $\sum_{i,i'} \kappa(S_{i,\cdot}, S'_{i',\cdot}) < 0$, there must be a $j$ and $j'$ for which $\varphi(S_{i,\cdot})_{j,j'} > 0$ but $\varphi(S'_{i',\cdot})_{j,j'} < 0$ (or vice versa), and thus the hypothesis test in Lemma 3 would reject the null hypothesis that $X_j \perp\!\!\!\perp X_{j'}$ but fail to reject that $X'_j \perp\!\!\!\perp X'_{j'}$. $\qquad\square$

**Corollary 11** Due to the relationship between independence structure and causal structure, an immediate of result of Theorem 10 is that $\sum_{i,i} \kappa(S_{i,\cdot}, S'_{i',\cdot}) < 0$ implies $X$ and $X'$ have different causal structures.

**Theorem 12** Let $d$ be the distance measure between unconditional equivalence classes of ancestral graphs over $m$ vertices, $d(\mathcal{U}, \mathcal{U}') = m^2 - |\{(j, j') : E^{\mathcal{U} \bullet \mathcal{U}'}(j, j') = \text{`}\leftrightarrow\text{'}\}| - m$. For given sample sets $S, S'$ (i.e., real $n \times m$ matrices), use $\bar{\varphi}(S)$ to denote the mean of the sample in kernel space, $\sum_i \varphi(S_{i,\cdot})$, and say $S \sim_K S'$ if and only if $\bar{\varphi}(S) \sim_O \bar{\varphi}(S')$; denote the corresponding quotient set by this equivalence class as $\mathbb{R}^{n,m} / \sim_K = \mathbb{K}^{n,m}$ and a representative from each equivalence class as $Q \in [S]$. Let $\delta$ be the distance between sets of samples in $\mathbb{K}$ defined as $\delta(Q, Q') = m^2 - \frac{1}{2n^2} \sum_{i,i'} \gamma(Q_{i,\cdot}, Q'_{i,\cdot})$. Let $b : \mathbb{U}^m \to \mathbb{K}^{n,m}, b : \mathcal{U} \mapsto \Omega$, where $\Omega$ is the unique element in $\mathbb{K}$ such that $\text{sign}(\bar{\varphi}(\Omega)) = a(\mathcal{U})$. Then $b$ is a distance-preserving map (i.e., an isometry) from the metric space $(\mathbb{U}^m, d)$ to $(\mathbb{K}^{n,m}, \delta)$.

*Proof.* Notice that $(\mathbb{U}^m, d)$ is indeed a metric space (Choudhary, 1993, Ch. 2): $d(\mathcal{U}, \mathcal{U}') = 0$ iff $\mathcal{U}^{-1} \bullet \mathcal{U}'$ is the empty graph, which happens iff $\mathcal{U} = \mathcal{U}'$; the symmetry of $d$ follows from the symmetry of $\bullet$; and for subadditivity of $d$, observe that for vertices $j, j'$ in arbitrary 2-vertex graphs $\mathcal{U}, \mathcal{U}', \mathcal{U}''$ we have either $d(\mathcal{U}, \mathcal{U}'') = 2$, in which case $d(\mathcal{U}, \mathcal{U}') + d(\mathcal{U}', \mathcal{U}'') = 4$, or we have $d(\mathcal{U}, \mathcal{U}'') = 0$, in which case $d(\mathcal{U}, \mathcal{U}') + d(\mathcal{U}', \mathcal{U}'')$ is either 0 or 4—in both cases $d(\mathcal{U}, \mathcal{U}'') \leq d(\mathcal{U}, \mathcal{U}') + d(\mathcal{U}', \mathcal{U}'')$; this easily extends to graphs of arbitrary numbers of vertices. Likewise, $(\mathbb{K}^{n,m}, \delta)$ is a metric space: $\delta(Q, Q') = 0 \iff \frac{1}{2n^2} \sum_{i,i'} \gamma(Q_{i,\cdot}, Q'_{i,\cdot}) = m^2 \iff \bar{\varphi}(Q)_{j,j'} = \bar{\varphi}(Q)_{j,j'}$, for all $j, j'$, so iff $Q = Q'$; symmetry and subadditivity of $\delta$ follow from the symmetry and subadditivity of $\gamma$.

Finally, to show $b$ is an isometry, we must show (i) that it is bijective and (ii) that for all $\mathcal{U}, \mathcal{U}' \in \mathbf{U}^m$, $d(\mathcal{U}, \mathcal{U}') = \delta(b(\mathcal{U}), b(\mathcal{U}'))$. For (i), observe that by the group isomorphism $a$ and definition of $b$, we have $\mathcal{U} \neq \mathcal{U}' \implies a(\mathcal{U}) \neq a(\mathcal{U}') \implies Q \neq Q' \implies b(\mathcal{U}) \neq b(\mathcal{U}')$ and so $b$ is injective. Also observe that because $\mathbb{K}$ is exactly the set of representatives of orthant equivalence classes of sample sets in kernel space, then for every $Q \in \mathbb{K}$, there exists a $\mathcal{U}$ such that $b(\mathcal{U}) = Q$, and so $b$ is surjective.

For (ii), isomorphism $a$ and the relation between element-wise product and Frobenius inner product allow us to write $d(\mathcal{U}, \mathcal{U}') = m^2 - \sum_{j,j'} (O \odot O')_{j,j'} = m^2 - \langle O, O' \rangle_F$. Substituting $O, O'$ with their corresponding $\Omega, \Omega'$, and because the Frobenius inner product is a sesquilinear form, we can write $d(\mathcal{U}, \mathcal{U}') = m^2 - \frac{1}{n^2} \sum_{i,i'} \langle \varphi(\Omega_{i,\cdot}), \varphi(\Omega'_{i,\cdot}) \rangle_F$, which by Definition 10 finally gives us that $d(\mathcal{U}, \mathcal{U}') = \delta(\Omega, \Omega')$, completing the proof. $\qquad\square$

In less formal terms, Theorem 12 shows how the space of unconditional equivalence classes of ancestral graph corresponds to the space of real matrices, which is a common space for samples to lie in. More specifically, it shows how the structure defined by distances between graphs is the same as the structure defined by distances between sets of samples and how this sample distance is related to our kernel $\kappa$. Note that this is much stronger than Theorem 10: not only can $\kappa$ tell us that two sets of samples come from different causal models, it gives a measure of just how different the causal models are, in terms of their differing unconditional nonlinear independencies/$m$-separation statements.

To summarize, we began by defining $\varphi$ (Definition 2), which maps a given data set into a new higher-dimensional feature space. This feature space corresponds to a space of causal graphical models, such that samples which are similar in the new feature space must come from similar causal models (Theorem 8). Our main contribution then is to propose the dependence contribution kernel $\kappa$ (Definition 9).This kernel $\kappa$ is guaranteed not only to tell us that two sets of samples come from different causal models (Theorem 10 and Corollary 11) but furthermore exactly how different the causal models are (Theorem 12), all without the computational expense of explicitly projecting samples or learning causal models. Thus, $\kappa$ is well-suited for addressing the causal clustering problem and ensures that resulting clusters will be structurally homogeneous so that subsequent causal structure learning will be more informative.

## 3 Application

We use kernel $k$-means with our dependence contribution kernel to cluster a gene expression data set and then use the measurement dependence inducing latent (MeDIL) causal model framework for structure learning within each cluster (Markham and Grosse-Wentrup, 2020). The goal of causal clustering here is to reason about the different latent transcription factor (TF) networks governing gene expression (see Verny et al., 2017; Hackett et al., 2020, for other latent causal model approaches to learning TF networks). The original data set comes from Iyer (1999) and can be found at `genome-www.stanford.edu/serum/data/fig2clusterdata.txt`, with subsequent analysis by Dhillon et al. (2003, 2004). All of the code for our analysis is open source and available at `https://non-anonymous-link.after-review`.

The data consists of the measured gene expression levels of 517 different genes from human fibroblast cells in response to serum exposure, measured at 11 different time points, i.e., there are 517 samples and 11 different features. In genetics applications, it is not unusual to consider genes to be samples and expression (over time) to be features—indeed the three previous analyses of this data all have this approach—and the intuition is simply that we wish to cluster genes based on patterns in their expression levels over time, in order to identify subsets of genes that are controlled by the same gene regulatory network. Also notice that such data exemplifies the structurally heterogeneous populations discussed in Section 1: different genes can of course be regulated by different TFs, and so we can better represent the data by first clustering it into subpopulations that are more homogeneous and then performing causal structure learning on each subpopulation.

For clustering, we used $k = 6$, which we found by looking at both the Variance Ratio Criterion (Caliński and Harabasz, 1974) and the Silhouette Coefficients (Rousseeuw, 1987), computed with the scikit-learn machine learning toolbox (Pedregosa et al., 2011). We implemented (unweighted) kernel $k$-means ourselves, using the pseudocode given by Dhillon et al. (2004), with initial mean points drawn uniformly at random from the sample set, and with significance level $\alpha = 0.1$ for the kernel parameter $\mathcal{T}(\alpha)$. We then used the MeDIL (Markham et al., 2020) package to learn the dependence structure and latent causal models for each cluster.

Figure 1 shows an example of our results for three of the six gene clusters: Figure 1a shows their distance covariance heatmaps and estimated nonlinear dependence structure with significance level $\alpha = 0.1$ (so the axes are the 11 different features, i.e. the time, in hours, at which gene expression level was measured), while Figure 1b shows their corresponding causal structures, with measurement variables $M_0-M_{10}$ for each of the features and learned latent variables $L$ for different posited TFs.

The results show a clear difference in causal structure for the different clusters and allow us to reason about the latent TFs regulating genes in different clusters: notice that the latents in cluster K1 each cause only two or three measurement variables that tend to be close together—e.g., $L_1$ causes $M_1$ and $M_2$, indicating the TF corresponding to $L_1$ is "short-acting", only affecting gene expression from 30 minutes ($M_1$) to 1 hour ($M_2$) after serum exposure; in contrast, the latents in cluster K3 each cause between two and seven measurement variables that tend to be more spread out—e.g., $L_1$ causes $M_1$ and $M_7$, indicating the corresponding TF is more complicated, "long-acting" but not

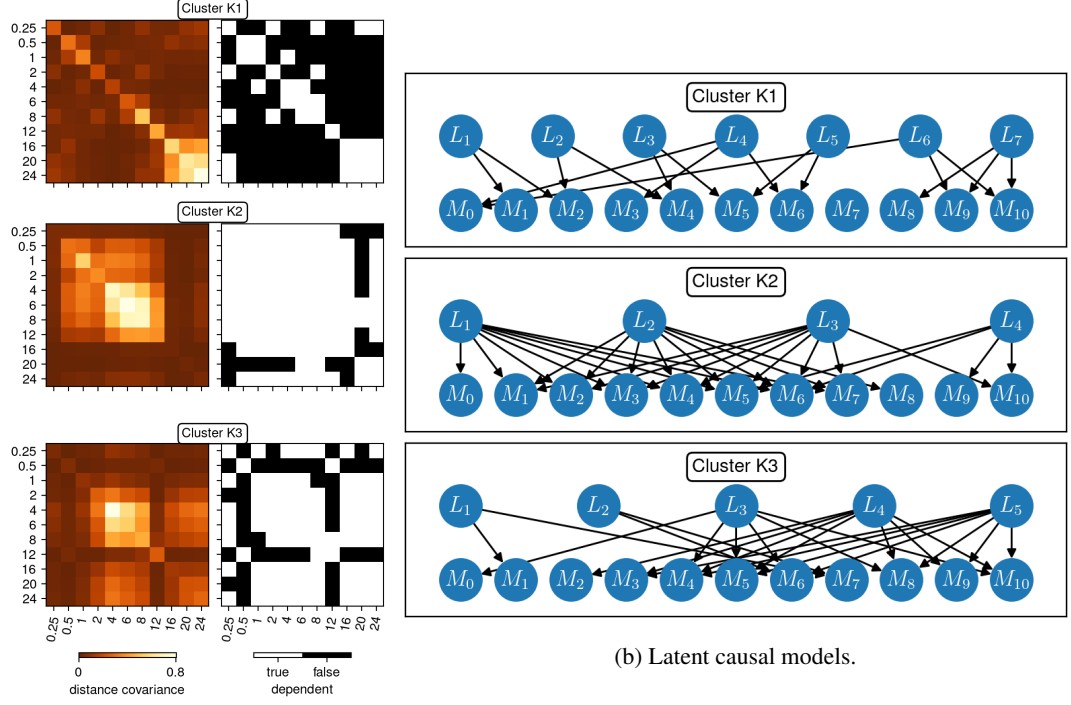

(b) Latent causal models.

(a) Dependence structures.

Figure 1: Results of dependence contribution kernel clustering with significance level $\alpha = 0.1$.

continuously so, affecting gene expression 30 minutes ($M_1$) and 12 hours ($M_7$) after serum exposure, but independently of gene expression in the time between.

Our results are especially noteworthy compared what happens if one ignores the heterogeneity of the data and learns a causal structure for the entire data set without first clustering with our kernel: in that case, all of the measurement variables are dependent, with a single latent causing all of them, and no meaningful conclusions can be drawn about how unmeasured transcription factors regulate measured gene expression, i.e., the heterogeneity obscures the underlying causal structures.

## 4  Discussion

We address the problem of causal clustering—that is, finding the different causal structures underlying a structurally heterogeneous data set. Our main contribution is to develop the *dependence contribution kernel* and prove its suitability for the causal clustering task. This allows us to first use the kernel with existing clustering methods, such as kernel $k$-means or DBSCAN, to identify homogeneous subpopulations. Then we use existing causal structure learning methods on each subpopulation. The kernel guarantees that each subpopulation is more structurally homogeneous and therefore the resulting causal structures better capture the causal structures within the data than if a single model were learned for the entire heterogeneous population.

Furthermore, we prove several interesting theoretical properties of our kernel, including (i) that it can be used as a statistical test for the hypothesis that two sets of samples come from different causal structures, as well as (ii) how it induces a metric space that is isometric to the one defined by Hamming distance between ancestral graphs, i.e., comparing sets of samples with our kernel is equivalent to first estimating the causal graphs of the different sets and then comparing those graphs. Beyond the practical applications of our kernel, as shown by our application in reasoning about latent transcription factor networks that regulate gene expression, this work also draws from and suggests further fruitful connections between a variety of fields, including causal inference, kernel methods, and algebraic statistics.

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
