# OpenReview forum: "A Distance Correlation-based Kernel for Nonlinear Causal Clustering in Heterogeneous Populations"
_NeurIPS.cc/2021/Conference — NeurIPS 2021 Submitted_

### Official Review · Reviewer_TQkh · 2021-07-16

**Rating:** 7
**Confidence:** 3

**Summary:**

Paper proposes a new distance covariance based kernel for causal clustering of heterogenous population data. It presents theoretical results  on how kernel it can be used as a two-sample statistical test for causal data and induces a metric space on ancestral graphs. Empirical results show leaned causal graphs for gene expression data.

**Ethical Concerns:**

No specific ethical concerns.

**Limitations And Societal Impact:**

Paper describes limitations of the causal structures learned and states assumptions clearly. Societal impact is not direct as it is mostly a theoretical work.

**Main Review:**

Idea of a distance covariance kernel for two-sample statistical testing of causal data is novel and interesting and is able to check for non-linear dependencies which follows from properties of distance covariance. Hilbert Schmidt Independence criterion (HSIC) which is an equivalent criterion of distance covariance could also used to defined a similar kernel on causal structures but not addressed on the paper.

Results on two-sample statistical testing and the induced metrics space on ancestral graphs provide a strong theoretical justification for the proposed kernel.

Empirical results on learned causal structures from gene expression data show the learned latent causal models but they still seems qualitative and verification on simulation data would be valuable here. Paper is well written and provides theoretically strong justification of the work which would be interested to the community.

**Time Spent Reviewing:**

4

---

> ### Author Response · Authors · 2021-08-09
> **We briefly respond to the reviewer's two main concerns, noting that our theory-based paper is intended to primarily address/interact with the causal inference community.**
>
> We are excited to see that you find our kernel idea "novel and interesting" and to have a "theoretically strong justification"!
> To your comment about HSIC: due to space constraints in the paper, we kept the Related Work section focused specifically on the causal inference literature instead of further discussion of the HSIC [e.g., Gretton et al. (2005, 2008) and Sejdinovic et al. (2013), lines 376, 378, and 419, respectively in the References] and other interesting work, such as graph kernel embeddings [e.g., Cai et al. 2018, line 355].
> And about our application: with our paper being theory-focused, we used the limited remaining space to show a real-world application to help motivate the theoretical problem and show just one example (out of many possible) of how our kernel can be used, with a particular data set, clustering method, and causal structure learning method; nevertheless, thank you for the helpful feedback that verification on simulation data would be valuable here!

---

### Official Review · Reviewer_jXQL · 2021-07-17

**Rating:** 6
**Confidence:** 3

**Summary:**

Authors proposed the dependence contribution kernel to resolve the homogeneous clusters in the projected feature space. This kernel is able to cluster samples based on difference in nonlinear independence structures.

**Limitations And Societal Impact:**

(1)	It would be helpful to perform a simulation study to demonstrate the power of the distance covariance-based kernel. The dataset (Iyer, 1999) is quite outdated as there are only 517 genes available. Nowadays it’s very common to have over 10,000 genes from the experiment. I’m curious how the kernel behaves when the number of genes scales up. The other possible source of heterogenous data is the whole-blood/PBMC gene expression as blood is a heterogenous collection of different blood cells. Maybe this kernel is helpful to discover the cell markers for different cell population which is an important question in biology.

(2)	Authors should at least include one baseline model/kernel, e.g., plain k-means with Euclidean distance to highlight the novelty of the distance covariance-based kernels.

(3)	There is ground truth based on biological knowledge regarding TF regulation. It’s impossible to make sense of the results without knowing what members are in each TF cluster. The other thing authors can try is to perform gene set enrichment analysis to make sense of the TF clusters.


**Main Review:**

The manuscript, especially the theory section, is well organized. Comparably the application section needs to be further polished.

**Time Spent Reviewing:**

1

---

> ### Author Response · Authors · 2021-08-09
> **We thank the reviewer for their helpful feedback about applications, while noting that the paper is theoretical, and so the limited paper length is devoted to rigorously presenting and proving theoretical results rather than more extensive empirical verification.**
>
> Thank you for the helpful suggestion about a potentially better data set and your valuable advice about other ways of improving our application section!
> Given the space constraints of this paper and our own research backgrounds, we focused on rigorously presenting our kernel and, in the words of another reviewer, providing "a strong theoretical justification for the proposed kernel".
> Thus, the comparatively smaller application section is meant more as a brief demonstration of our theoretical results in practice, to help provide motivation and context for the problem, as opposed to providing the sort of extensive practical justification of our kernel that would be possible with more space or in an application- rather than theory-focused paper.
> We also note that our results on this data set can be compared to those of (Dhillon et al. 2003, 2004), as cited in the beginning of our Application section, which analyses the same data set but using different existing kernels with kernel k-means.

---

> > ### Comment · Reviewer_jXQL · 2021-08-27
> > **Respond to authors’ rebuttal**
> >
> > I appreciate the efforts to formulate this work with a solid theoretic foundation. But I am not fully convinced by your response. If you feel the paper length is limited, you can incorporate more details into the supplementary instead of taking this as an excuse. The current presentation of application result doesn’t provide enough details to justify whether this kernel is able to help retrieve the correct interaction structures from the data. I don’t even know what L1,L2,…, M0, M1,… means which makes it impossible to make sense of the causal models (Figure 1b).
> >
> > The other concern is related to lacking proper comparisons with other related models. Without proper comparison with baseline models, it’s hard to evaluate the novelty of the method.

---

> > > ### Author Response · Authors · 2021-09-09
> > > **Clarification**
> > >
> > > Thank you for taking the time to consider and reply to our first response. We would like to clarify some of the points you mention, namely (i) the justification for the kernel and its use here, (ii) interpreting the causal models in Figure 1b, and (iii) the novelty of our method.
> > >
> > > (i) You write "The current presentation of application result doesn’t provide enough details to justify whether this kernel is able to help retrieve the correct interaction structures from the data". We emphasize that in the present application, the kernel is only used for clustering the data into structurally-homogeneous subpopulations. Theorem 12, which establishes that using our kernel to compute distances between samples is isometric to computing the distance between their generating causal ancestral graphs, is the theoretical justification for this use, and empirically, this can be seen in Figure 1a in that the different clusters found with our kernel have different structures. As for "retrieving the correct interaction structures from the data", this is the task of causal structure learning (not our kernel itself), and any number of causal structure learning algorithms could be used. We chose for this particular application to use the structure learning algorithm for the MeDIL causal model framework (Markham and Grosse-Wentrup, 2020), because it best aligned with the sassumption of our data consisting of meausurement variables, but other algorithms such as PC/IC, GES, FCI, etc. could be used just as well (if one accepts their corresponding assumptions).
> > >
> > > (ii) As we explained in lines 310--323, the variables M_0,...M_10 are the 11 different features of the data set, i.e., the gene expression levels at different times, while the variables L_1,... are the latent variables learned by the causal structure learning algorithm we used (which comes from MeDIL framework and is not part of our kernel or our contribution in this paper), and they can be interpreted as the latent transcription factors responsible for inducing the dependencies between the measured gene expression levels.
> > >
> > > (iii) The first paragraph of Section 3 mentions three other papers that have looked at this data. Our point in mentioning these was so that interested readers could look at their results (as a baseline) and compare them to ours. More importantly, however, the novelty of our approach is not simply that learned a different clustering for this data set than previous methods, or how this clustering leads to structure learning with MeDIL causal models---the novelty of our approach is in considering the problem of causal structural heterogeneity and providing a kernel with a solid theoretical foundation for addressing this problem, i.e., the novelty of our approach is in the idea of embedding nonlinear, causal ancestral graphical models into a kernel space so that applying the kernel to samples is guaranteed to provide a measure of similarity between their generating causal structures (all done implicitly and efficiently, thanks to the kernel trick).

---

### Official Review · Reviewer_vQoB · 2021-07-31

**Rating:** 4
**Confidence:** 4

**Summary:**

This paper focuses on the problem of causal structural learning from heterogeneous populations, where the label or index of the population is unknown. The main contribution of this paper is on developing a distance covariance-based kernel to measure the similarity between the underlying nonlinear causal structures of different samples, so that we can first perform clustering to identify the homogeneous subpopulations and then use existing methods to learn a causal structure for each of these subpopulations. The proposed method is tested on genetic data.


**Limitations And Societal Impact:**

yes

**Main Review:**

This paper studies an interesting problem—how to do causal structural learning when the data may be from different populations, without knowing their labels. However, overall, the writing needs to be polished to improve the readability, including the definitions and theorems.

My main concern of the proposed kernel metric is as follows. In Theorem 10 and Corollary 11, the authors show that if the kernel metric is negative, then X and X’ have different causal structures. The problem is that the theorem does not show the if and if only property; when the kernel metric is positive or zero, how should the conclusion be? Without the if and only if property, I do not think it can be applied to do clustering. Another issue is that Theorem 10 requires to know which n samples are in X and which n’ samples are in X’, and Theorem 10 holds asymptotically; but at the beginning of clustering, we need to start from n=n’=1, where the asymptotical result does not seem applicable.

Regarding the experiments, it would be more convincing if the authors can perform some simulation studies, since we can know the ground truth and estimate the accuracy of the proposed method. In addition, it is important to compare the results with different sample sizes and graph structures. For example, when some populations have very few samples, then how is the accuracy?

Update:

The author's response does explain some of my concerns about the if and only if property. However, I still have concerns about the writing and the experimental results. Also, the authors did not compare with any other methods.


**Time Spent Reviewing:**

4

---

> ### Author Response · Authors · 2021-08-09
> **We address each of the reviewer's concerns/questions, emphasizing that the isometry/bijection in Thrm. 12 (as opposed to the "if" in Thrm. 10 or Corr. 11) is the primary justification for our kernel to be used in clustering.**
>
> Thank you for the feedback about the readability, including of the definitions and theorems, and for your questions and concerns!
> We will specifically address each question/concern below, but in summary: our main contribution is a new kernel, including proofs (Thrm. 12) of how it defines a distance measure between samples in the kernel space that is isometric (thus there is a bijection, an "if and only if") to measuring distance between ancestral graph causal models (note that this is even stronger and a better justification for clustering than the kernel's use in statistical hypothesis testing, as in Thrm. 10 and Corr. 11), and our kernel is not tied to any specific clustering or causal inference method, all of which have their own associated assumptions and limitations.
>
> Now to each question/concern in more detail:
>
> -   "Theorem 10 and Corollary 11, the authors show that if the kernel metric is negative, then X and X’ have different causal structures. The problem is that the theorem does not show the if and if only property"
>
> &#x2013; For Thrm. 10, the "if" is because the error is bounded by the alpha parameter of the kernel (which is why the test uses a threshold), which it inherits from Lemma 3. We chose this presentation to better align with conventions in the statistics literature, but adding an "only if" is possible by computing an additional bound and specifying an interval instead of a threshold. Furthermore, instead of taking this approach to extend the hypothesis test to justify the use of the kernel in clustering, we chose to prove the isometry (i.e., bijection, so "if and only if") in Thrm. 12, showing that the distance between any two points in the kernel space corresponds exactly to the distance between their generating causal models in the space of ancestral graphs.
> &#x2013; Additionally, Corr. 11 has an "if" because it is stated with respect to particular causal structures&#x2014;if we instead change the statement to be about equivalence classes of causal structures, then we can add an "only if". Note that this is a standard limitation of causal inference from observational data, as discussed in the beginning of Section 2.2, and it is not unique to our kernel.
>
> -   "When the kernel metric is positive or zero, how should the conclusion be?"
>
> &#x2013; In this case, the conclusion should be "if we perform statistical hypothesis testing for independence on these two different sample sets using distance covariance, then they are not guaranteed (up to the error rate based on the alpha parameter) to have the same independence structure". This is perhaps a bit of a pedantic conclusion, in line with the statistics literature in which one either "rejects" or "fails to reject" (but does not accept) the null hypothesis.
>
> -   "Without the if and only if property, I do not think it can be applied to do clustering"
>
> &#x2013; The applicability of our kernel to clustering comes from Thrm. 12, which establishes an isometry, as opposed to the weaker Thrm. 10 and Corr. 11.
>
> -   "Another issue is that Theorem 10 requires to know which n samples are in X and which n’ samples are in X’ and Theorem 10 holds asymptotically; but at the beginning of clustering, we need to start from n=n’=1, where the asymptotical result does not seem applicable"
>
> &#x2013; This is handled differently by different clustering methods, but in general, clustering methods optimize the clusters according to some metric. The point of using a clustering method with our kernel is exactly to solve the problem of finding which points come from X and which come from X'. Importantly, clustering methods aren't using our kernel to perform hypothesis testing as in Thrm. 10&#x2014;they are rather using our kernel in a similarity/distance measure (which by Thrm. 12 is isometric to distance between generating ancestral graphs) to optimally find clusters of points from the same independence structures. After clustering, one can then perform causal structure learning (for which asymptotical results are standard), but this different from the clustering problem our kernel is helping to address via Thrm. 12.
>
> -   "It is important to compare the results with different sample sizes and graph structures. For example, when some populations have very few samples, then how is the accuracy?"
>
> &#x2013; Because of the isometry in Thrm. 12, the results for different graph structures are guaranteed to vary according to the distance between the graphs. The accuracy of clustering when some populations have very few samples depends on the specific clustering algorithm and parameters used, just as the accuracy would for other existing kernels, and thus it is important to use standard cluster evaluation methods, such as the Variance Ratio Criterion or the Silhouette Coefficients as we did in Section 3.

---

### Decision · Program_Chairs · 2021-09-27

**Decision:**

Reject

**Comment:**

This paper proposes a distance covariance-based kernel designed specifically for measuring the similarity between the underlying nonlinear causal structures of different samples, in order to deal with the problem of causal structural learning from heterogeneous populations without known labels, in which a single causal structure does not adequately describe all population members. According to the reviews, the paper would greatly benefit from simulation studies to demonstrate the power of the distance covariance-based kernel, and the authors should at least include one baseline model/kernel, e.g., plain k-means with Euclidean distance, to highlight the novelty of the distance covariance-based kernels.